# Xmrks the Spot: Fish Models for Investigating Epidermal Growth Factor Receptor Signaling in Cancer Research

**DOI:** 10.3390/cells10051132

**Published:** 2021-05-07

**Authors:** Jerry D. Monroe, Faiza Basheer, Yann Gibert

**Affiliations:** 1Department of Cell and Molecular Biology, Cancer Center and Research Institute, University of Mississippi Medical Center, 2500 North State Street, Jackson, MS 39216, USA; jmonroe1@umc.edu; 2School of Medicine, Deakin University, Locked Bag 20000, Geelong, VIC 3220, Australia; faiza.basheer@deakin.edu.au

**Keywords:** epidermal growth factor receptor, Xmrk, cancer, zebrafish, *Xiphophorus*, signal transduction, drug discovery, melanoma

## Abstract

Studies conducted in several fish species, e.g., *Xiphophorus hellerii* (green swordtail) and *Xiphophorus* maculatus (southern platyfish) crosses, *Oryzias latipes* (medaka), and *Danio rerio* (zebrafish), have identified an oncogenic role for the receptor tyrosine kinase, Xmrk, a gene product closely related to the human epidermal growth factor receptor (EGFR), which is associated with a wide variety of pathological conditions, including cancer. Comparative analyses of Xmrk and EGFR signal transduction in melanoma have shown that both utilize STAT5 signaling to regulate apoptosis and cell proliferation, PI3K to modulate apoptosis, FAK to control migration, and the Ras/Raf/MEK/MAPK pathway to regulate cell survival, proliferation, and differentiation. Further, Xmrk and EGFR may also modulate similar chemokine, extracellular matrix, oxidative stress, and microRNA signaling pathways in melanoma. In hepatocellular carcinoma (HCC), Xmrk and EGFR signaling utilize STAT5 to regulate cell proliferation, and Xmrk may signal through PI3K and FasR to modulate apoptosis. At the same time, both activate the Ras/Raf/MEK/MAPK pathway to regulate cell proliferation and E-cadherin signaling. Xmrk models of melanoma have shown that inhibitors of PI3K and MEK have an anti-cancer effect, and in HCC, that the steroidal drug, adrenosterone, can prevent metastasis and recover E-cadherin expression, suggesting that fish Xmrk models can exploit similarities with EGFR signal transduction to identify and study new chemotherapeutic drugs.

## 1. Introduction

The epidermal growth factor receptor (EGFR) is a member of the erythroblastic leukemia viral oncogene (ErbB) family of receptor tyrosine kinases (RTKs). Ligand-dependent EGFR activation triggers receptor autophosphorylation creating phosphotyrosine-binding sites for other proteins that then signal along various signal transduction pathways, e.g., mitogen-activated protein kinase (MAPK), phosphoinositide 3-kinase/protein kinase B (PI3K/Akt) and c-Jun N-terminal kinase (JNK), which control cellular behaviors, including adhesion, migration and proliferation [1,2] (Figure 1 and Figure 2). Mutations in EGFR have been linked to developing various diseases, including diverse cancers, including melanoma and hepatocarcinoma (HCC) [3,4]. Comparative genomic studies have shown that species of many other organisms, including fish, have genetic sequences homologous to EGFR [5].

Crosses between species of the fish genus *Xiphophorus*: *X. maculatus* (the southern platyfish) and *X. hellerii* (the green swordtail) produce fish that spontaneously develop malignant melanoma induced by a proto-oncogene encoding a receptor tyrosine kinase designated *Xiphophorus* melanoma receptor kinase (Xmrk), which is located on the sex chromosome [6]. The encoded Xmrk protein is structurally related to the human EGFR with an extracellular ligand-binding domain, a transmembrane domain and an intracellular catalytic domain [7]. Studies in human melanoma cell lines and the *Xiphophorus* cross-derived melanoma cell line (PSM-1) [8] using chimeric receptors combining the human epidermal growth factor receptor (HER) with Xmrk (HER-mrk) have shown that Xmrk displays constitutive autophosphorylation and is a functional receptor tyrosine kinase with activity in malignant melanoma [9]. Blockage of Xmrk’s kinase activity in *Xiphophorus* derived PSM cells demonstrates that receptor autophosphorylation is sufficient to trigger mitogenic signaling in melanoma cells [10]. Further, receptor stimulation induces autophosphorylation resulting in the production of the physiological ligand in an autocrine manner [11].

The role of Xmrk in melanoma also has been investigated in *Oryzias latipes*, the medaka fish. Ectopic expression of the Xmrk oncogene cloned from *Xiphophorus* and introduced into medaka embryos induces tumors, typically classifiable as cysts or hyperplasias, primarily in the epithelium, but also in the brain, retina, integument and pigment cells [12]. Expression of HER-mrk in medaka has demonstrated that one ligand, human transforming growth factor-alpha (hTGF alpha), can cause activation of this chimeric receptor [12]. Mutational studies of Xmrk in medaka embryos also have shown that point mutations can lead to constitutive, ligand-independent receptor kinase activity [13]. Medaka expressing HER-mrk, which exhibits invasive melanoma, has been used to show that activated Xmrk induces EGFR expression and ligand production leading to the formation of an autocrine loop that promotes pro-tumorigenic signaling [14].

RNA-Seq comparative analyses of *Xiphophorus* hellerii/maculatus hybrid fish and human melanoma tissues have shown shared pathways in these tissues governing: inflammation, cell migration, cell proliferation, pigmentation, cancer development, and metastasis [15]. Structural studies using the *Xiphophorus* and medaka models suggest that oncogenic signaling mechanisms and related translational studies can be studied using zebrafish transgenic Xmrk models. Recently, the tetracycline-controlled transcriptional activation Tet-on system [16,17] has been used in zebrafish to express Xmrk under the control of the liver-specific fatty acid-binding protein 10 (fabp10) promoter (also designated as liver fatty acid-binding protein (LFABP)) to study signaling in hepatocellular carcinoma (HCC) [18,19,20,21]. Here, we review the comparative genomic characteristics of the human EGFR and fish Xmrk receptor systems and the implications of this information for Xmrk oncogenic signaling in melanoma and HCC. We conclude by summarizing recent drug screening projects using fish models that have utilized Xmrk and the translational implications of these observations.

## 2. Genetic Characterization and Comparison of Xmrk with the Human EGFR

The Xmrk gene is hypothesized to have arisen from gene duplication and nonhomologous recombination during evolution [5,22]. There are three *Xiphophorus* orthologs of the EGFR gene: egfra, which encodes a tyrosine kinase receptor; egfrb, which is a proto-oncogene form termed INV-Xmrk; and ONC-Xmrk, a duplicated version of INV-Xmrk, which is highly expressed in melanoma where it is responsible for inducing oncogenic transformation [23,24,25,26,27]. Oncogenic activation of the ONC-Xmrk receptor occurs via two mutations, i.e., G336R and C555S, both located in the extracellular domain that is not found in INV-Xmrk; these mutations result in constitutive receptor dimerization leading to constitutive kinase signaling [27,28].

The Xmrk promoter is highly methylated in nontransformed tissues, but in the *Xiphophorus* cross-derived melanoma cell line (PSM-1) [8], it is unmethylated, suggesting that methylation status also regulates Xmrk expression [29]. Transcription of Xmrk does not stem from the action of cis-regulatory elements present in the promoter region of pigment cells, suggesting that epigenetic mechanisms also play a role in Xmrk’s oncogenic activation [30].

In comparison to Xmrk, the human EGFR is encoded by a single gene that has orthologs in all vertebrate species [5,31]; two transcriptions start sites have been identified, resulting in two major transcripts (10 and 5.6 kb) encoding the full-length receptor [31]. In addition to the two transcripts that encode EGFR Isoform A (the “full-length” or holo-receptor), three alternately processed transcripts encode alternate EGFR receptor isoforms (isoforms B, C, and D) [31], all of which are expressed in normal human tissues. While the full-length EGFR isoform (A) is known to signal through both ligand-dependent and independent signaling pathways [32], all three alternate isoforms lack the receptor kinase domain, rendering these isoforms unable to signal directly through protein phosphorylation. These isoforms have been hypothesized to function through signaling mechanisms similar to those used by the IGF-binding proteins, including regulating cell surface to extracellular matrix interactions, i.e., as matricellular proteins [33].

The transmembrane glycoprotein EGFR is derived from the 1210 residue precursor, which is cleaved at the N-terminus resulting in an 1186 residue mature EGFR protein [34]. The extracellular or ligand-binding domain (exons 1–16, amino acids 1–621) of the human EGFR receptor comprises four subdomains, two leucine-rich repeats (I and III) and two cysteine-rich repeats (II and IV) [35]. Subdomains I and III are involved in ligand-binding, domain II facilitates the formation of homo- or heterodimers with an analogous family member domain, and domain IV forms disulfide bonds to domain II. These subdomains are followed by a single hydrophobic transmembrane domain (exon 17, amino acids 645–668) and an intracellular cytoplasmic domain (exon 18–28, amino acids 669–1210). The cytoplasmic domain includes the tyrosine kinase (exons 18–24, amino acids 669–979) and C-terminal tail (exon 24–28, amino acids 980–1210) domains [35].

To date, several EGFR variants have been studied in humans [36,37,38]. In addition to the full-length isoform (isoform A), cells can also produce alternate EGFR isoforms that lack the intracellular domain [39]. The different roles of these alternative isoforms have not yet been established, but some have now been shown to play crucial roles [40]. Therefore, we analyzed the different EGFR transcript variants reported in the genomic and protein databases for zebrafish and humans, as summarized in Table 1 and Table 2.

Analysis of zebrafish genomic data version 11 on the Ensembl website, RGCz11; www.ensembl.org (accessed on 4 March 2021), revealed identifying six transcripts for egfra (egfra-201; 202; 203; 205; 206 and 207) (Table 1), and one transcript, which retained an intron and, therefore, is not listed in Table 1. Teleosts have experienced a fish-specific genome duplication [41], and, therefore, around one-third of the zebrafish genes are duplicated. In the zebrafish, a putative paralog named egfrb has been identified, but to date, no isoforms or transcript variants have been reported [42], and only one paper has reported the expression of the egfrb paralog in zebrafish [43]. The latest human genomic data on the Ensembl website (www.ensembl.org, accessed on 4 March 2021), human genome version (GRCh38.p13), revealed identifying 7 reported transcripts (EGFR-201 to 207) ranging from 1210 amino acids in length to 128 amino acids in length (Table 2). Ensembl human genomic data analysis also revealed the identification of two processed transcripts (561 and 452 bp, respectively) and one retained intron (665 bp), none of which are translated into a protein and, therefore, were excluded from Table 2.

NCBI analysis of the latest human EGFR transcript sequences (updated 7 March 2021) within the National Library of Medicine protein database revealed the existence of 9 EGFR transcripts encoding proteins from 1210 amino acids in length to 405 amino acids in length (Table 2). Comparison of the Ensembl and NCBI data sets revealed that 7 transcripts are common in both databases (highlighted in green in Table 2), while the reported transcript EGFR-205 is unique to the Ensembl database, and isoforms I, F and H are only present in the NCBI database (Table 2).

## 3. Xmrk in Fish Models of Melanoma

### 3.1. Tumor Suppression via Rab3d Signaling

Pigment cells in *Xiphophorus* platyfish and swordtail crosses were hypothesized to become malignant due to the loss of a platyfish regulatory gene, the tumor suppressor gene, R(Diff), which is normally present in non-hybrid parental fish where it prevents the activity of the Xmrk oncogene, Tu [44,45,46]. R(Diff) was originally identified as having a genetic sequence homologous to the mammalian cyclin-dependent kinase inhibitor 2 (CDKN2) gene family [47,48,49]. More recently, it has been determined that cdkn2ab is not the R(Diff) gene but is tightly linked to it and that the R(Diff) locus is *RAB3D*, a Ras-related small G protein with GTPase activity that regulates exocytosis [50]. In human HCC tissues, RAB3D can interact with Golgi membrane protein 1 (GOLM1), a protein that facilitates EGFR cycling from the Golgi apparatus to the plasma membrane, causing increased downstream oncogenic kinase signaling [51]. This recent observation suggests that RAB3D may regulate EGFR cycling, thereby reducing the number of active receptors on the cell surface, leading to decreased oncogenic EGFR signaling [50]. Further, RAB3D can modulate the secretion of matrix metalloproteinase-9 (MMP-9) from activated macrophages, facilitating migration into infected tissues [52]. Therefore, in fish, RAB3D also may promote antitumorigenic functions by recruiting immune cells to the tumor microenvironment. In contrast, upregulation of MMP-9 in breast cancer and other cancers enhances tumor cell migration and invasion [53]. Studies to further characterize the mechanistic roles played by RAB3D in Xmrk-driven melanoma signaling and parallel studies to examine the potential role of RAB3D in human melanomas are clearly warranted.

### 3.2. Signal Transducer and Activator of Transcription 5 Signaling

Stimulation of HER-mrk with EGF in PSM cells induces tyrosine phosphorylation, nuclear translocation and DNA binding of signal transducer and activator of transcription 5 (STAT5) followed by increased expression of the STAT5 target genes, cytokine-inducible SH2-containing protein (CISH), oncostatin M (OSM), and proto-oncogene serine/threonine-protein kinase PIM-1, whereas STAT1 and STAT3 signaling are not activated [54,55] (Figure 1). Expression of HER-mrk in Ba/F3 hematopoietic murine cells also modulates growth and survival signaling via STAT5. It induces the expression of the genes, proto-oncogene serine/threonine-protein kinase Pim-1 (pim-1), B-cell lymphoma (Bcl-x) and cellular myelocytomatosis oncogene (c-myc) [56] (Figure 1). Further, expression of HER-mrk in mouse melan-a cells treated with EGF caused activation of STAT5 and increased expression of Bcl-XL [57] (Figure 1). Studies in transgenic medaka have shown that expression of Xmrk is correlated with STAT5 activation and increased expression of microphthalmia-associated transcription factor (MITF), which upregulates B-cell lymphoma 2 (Bcl-2) and surviving, and is also associated with melanoma progression [58]. Interestingly, EGFR/STAT signaling may differ in human melanoma versus *Xiphophorus* Xmrk models since inhibition of c-Src kinase activity, but not EGFR, in human melanoma-derived cell lines, downregulates STAT3 signaling and Bcl-xL expression [59].

In human melanoma cells, EGF treatment can promote STAT5 activation via tyrosine-protein kinase Src (Src) and Janus kinase 1 (JAK1) signaling leading to STAT5 nuclear translocation along with upregulation of the STAT5 target, Bcl-2, an antiapoptosis protein [60]. Further, EGFR expression is upregulated in some human melanoma cell lines, and inhibition of STAT5 causes increased apoptosis [60]. CISH downregulates interleukin 15 (IL-15) signaling in natural killer cells, and Cish-/- mice injected with murine melanoma cells exhibit greatly reduced metastasis [61]. OSM is a cytokine belonging to the IL-6 family, and in the A375 human melanoma cell line, OSM receptor signaling is mediated by JAK1, JAK2, and tyrosine kinase 2 signaling via STAT3 and STAT5b [62]. Administration of interleukin-2 (IL-2) in patients with metastatic melanoma has shown that some immune cell populations exhibit elevated levels of activated STAT5 correlated with increased transcription of CISH and PIM-1 [63]. Expression of PIM-1 in human melanoma cell lines and tissue samples promotes cell migration, invasion and epithelial-mesenchymal transition (EMT), which can be suppressed by the microRNA, miR-542-3p [64].

MITF induces Bcl-2 expression, reducing tumorigenesis in human melanoma cells and increasing the tyrosinase expression associated with melanogenesis [65]. Constitutively activated human STAT5 causes enhanced expression of the antiapoptosis protein, Bcl-XL, promoting proliferation and survival in human melanoma cell lines [57]. A human A375 cell line melanoma xenograft murine model has shown that deletion of glucose-6-phosphate dehydrogenase (G6PD), which catalyzes the pentose-phosphate pathway, upregulates the antiapoptosis proteins Bcl-2 and Bcl-xL, whereas the tumor suppressor, Fas, was shown to be downregulated [66]. Downregulation of Bcl-2 and STAT1 in human melanoma cell lines also can prevent activation of STAT3 and 5 [67]. Although c-myc regulation of EGFR/STAT5 signaling has not yet been characterized in melanoma, in breast cancer cell lines expressing the estradiol receptor, EGFR and c-Src kinase function are correlated with estradiol-induced cyclin D1 and c-myc gene transcription [68]. Survivin-mediated signaling integrating EGFR/STAT5 also has not been studied in melanoma, but in A549 non-small cell lung cancer cells, drug inhibition studies decrease EGFR, JAK2, STAT3, and STAT5 activation, and also downregulated Bcl-2 and survivin preventing proliferation and promoting apoptosis [69].

### 3.3. Phosphoinositide 3-Kinase Signaling

The p85 adaptor subunit of phosphoinositide 3-kinase (PI3K) can bind to the autophosphorylated Xmrk receptor enabling PI3K to phosphorylate its downstream target, protein kinase B (Akt), in melanoma cells [70] (Figure 1). In *Xiphophorus*, the PI3K p85 subunit also can associate with the Src family non-receptor tyrosine kinase, p59fyn (Xfyn), which acts not only as an adaptor but also as an activator of PI3K [71] (Figure 1). Stimulation of Xmrk promotes Xfyn activity, causing the formation of a complex with focal adhesion kinase (FAK), which alters FAK activity and induces the formation of stress fibers, focal adhesions and pigment cell migration [72,73,74] (Figure 1). Mammalian Fyn also interacts with FAK suggesting that Fyn may play a role in human melanoma progression [75]. In the immortalized human keratinocyte cell line, HaCaT, H-Ras induces upregulation of Fyn via PI3K/Akt signaling, promoting increased cell migration and invasion by Fyn modulating FAK. However, activation of EGFR was not necessary to promote signaling via Akt [76], which suggests that some human melanomas may exhibit different signaling characteristics than in fish models of the disease. Alternatively, in BRAF inhibitor-resistant human melanoma cell lines, there is pronounced hypomethylation of the EGFR gene promoter, leading to increased expression of EGFR and metastasis through PI3K/Akt signaling [77]. Further, inhibition of BRAF causes upregulation of the EGFR/PI3K/Akt pathway while preventing action through the MEK/ERK pathway [77]. Together, these results suggest that the necessity for EGFR activation in PI3K human melanoma signaling may depend on the specific melanoma subtype and that signaling homologies between Xmrk fish melanoma models and human skin cancer may not always occur.

The *Xiphophorus* forkhead transcription factor, FoxO5, when phosphorylated by Akt, promotes melanoma proliferation, and forkhead transcription factor signaling in human melanoma is associated with cell cycle arrest and/or apoptosis through the PI3K/Akt pathway [78]. Expression of the transcription factor, forkhead box D3 (FOXD3), in human melanoma cell lines, is induced during blockage of the BRAF/MEK/ERK pathway leading to upregulation of ERBB3 and activation of Akt signaling, which promotes melanoma progression [79]. Although some human melanomas may signal primarily via PI3K/Akt instead of MAPK signaling, loss of the tumor suppressor PTEN can lead to MAPK pathway activation via translocation of β-catenin to the nucleus, where it acts to regulate MITF transcription and promote metastasis independently of PI3K/Akt signaling [80].

### 3.4. Mitogen-Activated Protein Kinase Pathway Signaling

The mitogen-activated protein kinase (MAPK) pathway incorporates kinases, e.g., MEK and ERK, to modulate cancer via EGFR signaling [81]. Growth factor receptor-bound protein 2 (GRB2) binds to Xmrk directly in PSM cells and also indirectly via the adaptor protein, Src homology and collagen (Shc), which allows the recruitment of the small GTPase, Ras, and activation of mitogen-activated protein kinases 3 and 1 (ERK1/2) [82,83] (Figure 1). In human melanoma tissues, mutations in Ras and its immediate downstream target, B-Raf, are associated with activation of ERK1/2, while wild-type Ras and B-Raf samples express much lower levels of activated ERK1/2 [84].

Stimulation of melan-a HER-mrk cells with EGF activates MAPK signaling and upregulates the adhesion protein, osteopontin (OPN), promoting melanocyte adhesion to collagen and preventing collagen-mediated apoptosis [85] (Figure 1). OPN also is upregulated in several human melanoma cell lines after EGF stimulation [86]. OPN binding to integrin activates nuclear factor kappa-light-chain-enhancer of activated B cells (NF-κB)-mediated signaling, which modulates the MAPK pathway and upregulates MMP-9, causing increased cancer cell migration and invasion [87,88,89,90].

In melanoma cells, Xmrk downregulates the tyrosinase gene, which controls melanin synthesis via MAPK inhibition of the transcription factor, microphthalmia-associated transcription factor (MITF), and suppresses melanocyte differentiation [91,92] (Figure 1). Similarly, in mouse and human melanocytes, B-Raf acts to suppress MITF signaling via ERK and promote proliferation, while upregulation of MITF prevents proliferation through the cAMP pathway [93]. In human melanocytes, MAPK signaling causes reduction of MITF and increases ERBB3, FOXD3 and NRG1 transcription factor gene expression, with NRG1 and FOXD3 also further upregulating ERBB3, which can heterodimerize with EGFR causing melanoma metastasis [94,95,96]. NRG1/ERBB3 signaling also can prevent human melanocyte maturation and promote proliferation and undifferentiated, migratory features [97].

HER-mrk can also activate p59fyn and prevent the MAPK phosphatase, MKP-1, from suppressing MAPK, promoting mitogenic signaling and preventing differentiation in melan-a cells [98] (Figure 1). In human melanoma cell lines, the pattern recognition receptor, RIG-1, can modulate MKP-1 and suppress cell proliferation by inhibiting MAPK signaling [99]. Further, human melanoma cell lines expressing the CD133 antigen can activate the PI3K/Akt pathway and suppress MAPK via MKP-1 signaling [100].

### 3.5. Chemokine Signaling

Studies in transgenic medaka melanoma models have shown that expression of Xmrk is correlated with increased gene expression of the chemokine, stromal-derived factor (SDF-1), which signals through the C-X-C chemokine receptor type, CXCR7, but not CXCR4, and is responsible for regulating melanoma progression [101]. In human epidermal melanocytes, SDF-1 was found to regulate cell migration via CXCR7, but not CXCR4, via the MAPK/ERK1/2 pathway mediated by β-arrestin 2 [102]. However, the signaling mechanisms active in normal human melanocytes may not be the same as those in melanoma cells. Some human melanoma cell lines do not express the CXCR7 receptor, but cell lines expressing only the CXCR4 receptor demonstrate SDF-1 modulated regulation of melanoma proliferation and migration [103]. A study conducted using exosomes from cultured melanoma cells has shown that CXCR7 expression was required to chemotactically migrate towards SDF-1 gradients [104]. In ovarian cancer, inhibition of the EGFR receptor has been shown to downregulate SDF-1α and ERK1/2 activation via the CXCR4 receptor causing reduced cell proliferation [105]. To date, SDF-1/CXCR7 signaling via EGFR has not been established in a cancer model. Therefore, additional research is required to properly characterize the EGFR/SDF-1/CXCR signaling axis in melanoma.

### 3.6. Regulation of the Extracellular Matrix

Gene expression analysis in the medaka mitf::xmrk model has shown that genes modulating the extracellular matrix (ECM) in early and advanced melanoma, including members of the laminin gene family and the integrin, α2β1, are upregulated. In contrast, several members of the collagen gene family are downregulated [106]. Expression of the HER-mrk transgene in melanocytes in the presence of collagen I matrix demonstrates that during EGF stimulation, migration is associated with SRC kinase function and independent of PI3K, MAPK and MMP signaling [107] (Figure 1). A study of cellular migration on collagen has shown that the drug casticin reduces human melanoma cell migration by downregulating the EGFR/Ras/ERK pathway, NF-κB and the matrix metalloproteinases, MMP-1 and -2, which promote cancer invasion by enzymatically degrading collagen in the basement membrane [108]. Laminin-2 is upregulated in metastatic human melanoma cells, and α2β1 integrin is the receptor that mediates laminin-2 adhesion [109]. Further, the interaction of laminin-2 with α2β1 integrin causes secretion of collagenase, leading to activation of the epidermal growth factor family member, p185/C-Erb B2 (HER2/neu) [109]. In some human melanoma cell lines, EGFR signaling acts through the cytoskeletal protein, filamin A (FLNa), a regulator of the PKB/Akt and ERK1/2 pathways, and is associated with migration of the integrin, α1β1, to focal adhesions where it promotes cell adhesion and migration through contact with collagen type I [110]. The results obtained from medaka and HER-mrk models suggest that Xmrk can regulate laminin and integrin-based mechanisms in human melanoma, but there may be mechanistic differences, e.g., in MAPK and MMP signaling that require additional characterization. In addition, the role of Xmrk in collagen signaling, both in terms of its mechanism of action and regulation of collagen levels in the ECM during melanoma cell migration, requires further elaboration.

### 3.7. Oxidative Stress Mechanisms

In *Xiphophorus* malignant melanoma tissue, oxidative stress increases and is correlated with upregulation of the antioxidative proteins, glutathione-S-transferase mu3 (GST mu3) and peroxiredoxin-6 (PRDX6), but not PRDX2, which was subject to a non-significant increase [111]. GST mu3 is an enzyme expressed in the skin that detoxifies oxidative stress products generated from exposure to UV irradiation [112]; however, its signaling characteristics in melanoma concerning EGFR have not yet been characterized. Reduced PRDX2 expression in human melanoma cells is associated with increased proliferation and migration via PRDX2 suppression of Src/ERK signaling, which also increased E-cadherin and retention of β-catenin at the adherens junction [113]. PRDX6 is upregulated in human melanoma cells, increases cell proliferation by activating Src signaling via the production of arachidonic acid, and its expression depends on EGFR signaling [114]. As PRDX2 regulation of oxidative stress may be different in *Xiphophorus* compared to human melanoma, this could mean that Xmrk signaling may not integrate PRDX2 function. However, there may be a compensatory relationship between PRDX2 and 6 in *Xiphophorus* melanoma that has yet to be characterized or that these antioxidant proteins respond to different degrees of oxidative stress.

### 3.8. MicroRNA Signaling

A study of miRNA expression and target gene regulation in the *Xiphophorus* and transgenic medaka Xmrk models have shown that in fish melanomas, miR-17, miR-18a, miR-20a2, miR-92a1, miR-126, miR-182, miR-210 and miR-214 were upregulated, and their target genes, RUNX1, HIF1A, TGFBR2, THBS1 and JAK2, were downregulated [115]. Further, miR-125b is downregulated in fish melanomas, and its target genes, ERBB3a and ERBB3b are upregulated [115].

MiR-20a2 and miR-92a1 have not yet been characterized in human skin cancer, but in human melanoma cells, miR-17 is overexpressed and acts to reduce expression of the posttranscriptional RNA-editing enzyme, ADAR1, leading to greater cell proliferation [116];. However, in a transgenic mouse model of miR-17, overexpression of the microRNA is associated with reduced tumor migration and decreased STAT3 signaling [117] (Figure 1). Upregulation of miR-18a, which is sponged by the long non-coding RNA (lncRNA), CASC2, promotes cell proliferation, migration, and invasion and suppresses ephrin receptor A7 mediated apoptosis [118,119] (Figure 1). Upregulation of miR-182 promotes migration and survival in human melanoma cell lines by preventing the transcription factors, MITF-M and FOXO3, and downregulates Bcl-2, cyclin D2, c-Met, the Akt and ERK1/2 pathways while increasing β-catenin expression [120,121,122] (Figure 1). Hypoxia upregulates miR-210 in human melanoma cells. It targets MNC, a transcriptional repressor of the oncogene c-Myc, and promotes its function in hypoxic conditions to increase melanoma proliferation and survival and can also act as an immunosuppressant against cytotoxic T lymphocytes [123,124]. MiR-214 is upregulated in melanoma, where it enhances metastasis and survival by decreasing expression of the integrin encoding gene, ITGA3, which encodes a portion of the integrin complex, and the tumor suppressor homolog, TFAP2C [125] (Figure 1).

Unlike miR-17, 18a, 182, 210, and 214, whose increased expression promotes melanoma, upregulation of miR-126 in human cells has anti-melanoma function through its modulation of v-crk sarcoma virus CT10 oncogene homolog (CRK), a regulator of migration and adhesion [126,127] (Figure 1). MiR-126 is suppressed by the lncRNA, LINC00888, and targets two metalloproteases, ADAM9 and MMP7 [126,127]. In B16F10 melanoma cells, increased miR-125B acted to suppress tumor cell migration by repressing Stat3 and promoting E-cadherin, while FAK acted to prevent the function of miR-125b [128] (Figure 1). Although regulating miRNAs in fish Xmrk melanoma models appears to be generally similar to their human counterparts, at this time, it is uncertain if the miRNAs modulated in fish target the same pathways or are similarly integrated into larger signaling networks. Further, as upregulation of miR-126 in human melanoma is associated with anti-cancer effect, some miRNAs may have different roles in humans as opposed to fish-based models. In addition, as the relationship between EGFR and these miRNAs in human melanoma has not yet been characterized, this suggests a future focus for research that could assist in developing fish-based translational Xmrk models.

## 4. Xmrk in Fish Models of Hepatocellular Carcinoma

### 4.1. Xmrk Single Transgene Studies

As the EGFR and its ligands are typically upregulated in HCC [3], knowledge obtained from highly tractable fish models expressing Xmrk could be invaluable in understanding the role of the EGFR receptor in liver cancer. To this end, transgenic zebrafish have been constructed that express Xmrk driven by a liver-specific fabp10 promoter which under doxycycline treatment induces HCC in juvenile and adult zebrafish [18]. This model demonstrated that during tumor induction, there was increased cell proliferation and activation of the downstream Xmrk targets, ERK and STAT5, while during tumor regression, which occurs upon removal of doxycycline, phosphorylation of ERK and STAT5 decreased [18] (Figure 2). Further, fabp10 promoter-driven transgenics analyzed with RNA-seq using Gene Ontology and Kyoto Encyclopedia of Genes and Genomes databases have shown that Xmrk-induced zebrafish HCC gene expression is similar to human HCC subtype, S2, which is characterized by enhanced Myc signaling, phosphorylated-ribosomal protein, S6, and epithelial cell adhesion molecule regulation [19] (Figure 2). In human cell lines and mouse xenografts, inhibition of p70S6 kinase activates S6 while reducing HCC growth and EMT [129]. Activation of S6 also caused downregulation of Slug, Twist and increased E-cadherin expression by suppressing FAK/Akt signaling [129] (Figure 2). MiR-146a targets the EGFR, and its expression in human HCC cell lines downregulates EGFR, ERK1/2, and STAT5 [130]. Therefore, Xmrk transgenic HCC zebrafish models, as in human HCC, modulate mechanisms integrating the ERK and STAT5 pathways. They may also signal through analogs to human Myc and p70S6 kinase and act on their respective downstream targets.

Studies of sex differences in Xmrk single transgenic HCC fish have revealed that male HCC tumors have higher serotonin and cortisol expression. They also have higher numbers of neutrophils and macrophages than female HCC tumors, suggesting that these signaling molecules may have a sex-based role in liver tumors and associated immune cell proliferation [131]. A relationship between either serotonin or cortisol signaling with EGFR and HCC has not yet been reported. However, serotonin treatment promotes HCC in human tissue samples via an mTOR-independent pathway that integrates p70S6 kinase and 4E-BP1 signaling [132]. Serotonin, EGFR and EGF levels are elevated in human cholangiocarcinoma cell lines, and even though serotonin had no effect on EGFR expression or activation or EGF levels, serotonin synergizes with EGFR to produce greater activation of the matrix metalloproteinase, MMP-9 [133]. When treated with cortisol, transformed malignant human mammary cells have increased EGFR and reduced MHC class-I molecules [134]. Increased cortisol levels in HCC patients are also correlated with reduced p53 expression [135]. Thus, zebrafish models suggest that there could be a sex-based role in the progression of human HCC via EGFR signaling and may indicate a future focus of HCC study on neurotransmitter or hormone-related mechanisms.

### 4.2. Multi-Oncogene Studies Integrating Xmrk

As the action of other oncogenes may alter the characteristics of Xmrk signaling alone, researchers have constructed multi-oncogene expressing transgenics to analyze the interaction of Xmrk with other cancer genes. Gene regulatory comparisons of transgenic zebrafish expressing Xmrk, Kirsten rat sarcoma virus (Kras) or Myc alone have shown that there is only limited overlap between these three genes and confirmed that Xmrk oncogene-induced zebrafish liver tumors were correlated with a small subset of human HCC samples [20]. Kras is a member of the Ras family of proto-oncogenic GTPases, which regulate MAPK/ERK1/2 and PI3K/Akt signaling and are frequently mutated in HCC patients exposed to vinyl chloride [136,137]. The proto-oncogene, c-Myc, is a transcription factor regulated by tumor growth factor-beta 1 (TGFβ1) and NF-κB in HCC that can be phosphorylated and activated by ERK1/2 in the MAPK signaling pathway, and whose expression is elevated in HCC tissues [138,139,140].

The Xmrk transgenic exhibited dysregulation of pathways involved in evading growth suppressors and immune destruction, cell cycle activation, increased RNA transcription, and proteasome function, and altered immune system characteristics [20]. Further, the Kras transgenic had upregulated EGFR, RAF/MEK/ERK and PI3K/Akt/mTOR signaling and glycogen synthase kinase 3 (GSK3) expression, which was associated with proliferation. At the same time, the Myc transgenic exhibited increased regulation of translation and proteolysis genes associated with growth suppression and the VEGF pathway [20]. However, crosses between the Xmrk and Myc single transgenic zebrafish (Xmrk/Myc) had more severe HCC than single transgene fish and enhanced glycolytic gene function associated with the Warburg effect [21]. The EGFR in human HCC is associated with the extracellular membrane protein, VersicanV1, a chondroitin sulfate proteoglycan, which can induce EGFR signaling via the PI3K/Akt pathway and promote the Warburg effect by shifting cellular metabolism from aerobic oxidation to lactic acid fermentation [141]. These studies suggest that an accurate analysis of Xmrk’s role in HCC using zebrafish models may consider the role of multiple oncogenes to appropriately assess signal transduction network function.

Men have a much higher incidence rate of HCC than women. Administering estrogen to male mice reduces the proinflammatory cytokine IL6 and decreased HCC incidence in carcinogen-exposed animals [142]. However, increased estrogen receptor (ER) concentration has been associated with tumor progression, and synthetic estrogens can increase EGFR expression and EGF binding [143]. Further, rapid estrogen signaling through a variant of ER has been identified as a mechanism that modulates HCC through EGFR/Src/ERK signaling during estrogen treatment in human HCC cell lines [144]. Studies in zebrafish double transgenics have revealed sex-based differences in HCC progression due to hormonal signaling. Male Xmrk/Myc double transgenics develop HCC faster than females, and treatment of these fish with the androgen, 11-ketotestosterone, promoted HCC proliferation, while treatment with the estrogen, 17β-estradiol, reduced HCC progression [145]. The effect of androgen signaling via EGFR has not been characterized in HCC. However, in breast and prostate cancer, EGF induces the formation of a complex between the EGFR, androgen receptor (AR), estrogen receptor (ER) and Src kinase [146]. Studies in knockout mice have shown that HCC development depends on AR expression independent of 5α-dihydrotestosterone levels and possibly acts to promote HCC through increased oxidative stress and DNA damage with suppression of the p53 signaling pathway [147]. As the results obtained with mammalian models have shown that androgen and estrogen signaling in HCC is complex and not necessarily consistent with those obtained with transgenic zebrafish models, zebrafish models of Xmrk modulated HCC might exhibit compensatory effects from transgene expression in zebrafish liver tissue or different expression patterns of ARs and ERs or distinct hormone responses compared to mammalian systems.

Elevated expression of the transcription factor, Twist, a regulator of EMT, occurs in metastatic human HCC samples and correlates with decreased E-cadherin expression [148]. Inhibition of hydroxysteroid (11-beta) dehydrogenase1 (HSD11b1), a prognostic biomarker of human HCC, in crosses between the tamoxifen-controllable Twist1a-ERT2 and Xmrk transgenic HCC zebrafish, suppressed the metastasis of transplanted human cell lines and led to recovered E-cadherin expression via Snail and Slug transcription factor signaling [149,150]. E-cadherin can bind to EGFR and regulate its activation leading to either anti- or pro-tumorigenic outcomes depending on the stage of HCC progression and the status of Wnt/β-catenin signaling [151]. In human HCC cell lines, the tyrosine kinase, c-Src, with EGFR, can downregulate E-cadherin, associated with increased Slug levels through the NF-κB pathway and HCC progression [152]. Further, hepatocellular carcinoma-related protein 1 (HCRP1), a protein responsible for degradation of ubiquitinated membrane receptors, is downregulated in HCC and correlated with increased EGFR activation, EMT, Snail and Twist1 expression [153]. Although a comparison of zebrafish Xmrk/Twist transgenic and human HCC might reveal similar signaling patterns integrating the Snail/Slug and E-cadherin pathways, the zebrafish model may not precisely recapitulate human HCC signaling at different stages of progression.

## 5. Translational Xmrk Studies

Fish models have been extensively used to screen chemotherapy drug candidates targeting a wide range of cancers, including melanoma and HCC [154,155,156,157]. Zebrafish embryo-based platforms have shown that the small molecule, SKLB226, can downregulate MITF mRNA, prevent zebrafish pigment cell migration and had similar effects on MITF and migration in mammalian melanoma cell culture [158]. MITF can regulate the expression of CDKN1A, an inhibitor of the cell cycle, suggesting that SKLB226 prevents MITF from blocking the action of CDKN1A [158]. Recently, a human melanoma cell xenotransplant zebrafish model demonstrated that the lectin, BEL β-trefoil, prevented melanoma migration and reduced expression of Runx2, a gene upregulated and associated with melanoma invasion and migration, which acts to regulate the tumor suppressor p53 in pathways integrating Ras/MEK/ERK and PI3K/Akt signaling [159]. Zebrafish embryonic and transgenic adult methods have also shown that the tyrosine kinase inhibitors, 419S1 and 420S1, have comparable or superior effects against HCC angiogenesis and migration than the FDA-approved broad-spectrum protein kinase inhibitor HCC drug, sorafenib, and lower toxicity, with 419S1 and 420S1 having a suppressive effect on the cell cycle genes, ccne1, cdk1, and cdk2, as did sorafenib [160]. Further, when zebrafish embryos with transplanted mammalian HCC cells were treated with the antioxidant propyl gallate, there was a significant reduction in HCC proliferation [161]. Although propyl gallate acts as an antioxidant in food preservative applications, in HCC, it acts to increase reactive oxygen species and superoxide levels, induces autophagy, and promotes apoptosis by increasing the expression of the proapoptotic proteins, Bax, Bad, cleaved PARP and caspase-3, while decreasing expression of the antiapoptotic protein, Bcl-2 [161].

The success of zebrafish models in drug screening applications suggests their utility in discovering new melanoma and HCC drugs. However, few studies have been conducted investigating the effect of drugs against cancers that incorporate Xmrk signaling in *Xiphophorus*, *Oryzias* and *Danio* models. A study conducted in PSM cells and *Xiphophorus* melanoma tissue samples has shown that Xmrk can activate the PI3K/Akt pathway, and inhibiting PI3K in PSM cells using Wortmannin and LY294002 prevented entry into the S-phase [70]. Another study in a medaka transgenic overexpressing Xmrk under the control of a pigment cell-specific MITF promoter utilized NanoString nCounter screening to quantify gene expression and characterize transcriptional disease signature (TDS) profiles after treatment with various drugs [162]. This analytical system identified transcriptional responses modulated by the melanoma drug trametinib, a MEK1 and MEK2 inhibitor, and the FDA-approved anti-cancer drug, cisplatin, on a large set of oncogenes suggesting that this system can be effectively used to screen candidate chemotherapy agents [162]. These results support the interpretation that trametinib may prevent Xmrk signaling via the Ras/Raf/MEK/ERK pathway, while cisplatin’s action likely does not integrate Xmrk signaling, as this platinum-based complex works by crosslinking DNA leading to DNA damage and induction of apoptosis [162]. Another study used a cross between a tamoxifen-controlled Twist1a-ERT2 transgenic zebrafish line and an Xmrk transgenic zebrafish that develops metastatic HCC to screen chemotherapy drugs. They found that the drugs adrenosterone, rabeprazole, and olmesartan, suppressed the metastasis of transplanted human liver cancer cells [150]. Adrenosterone is an inhibitor of hydroxysteroid (11-beta) dehydrogenase 1, which is highly expressed in metastatic HCC, rabeprazole acts to block hydrogen/potassium-transporting ATPase, and olmesartan prevents the action of angiotensin 2 types 1 receptor, which promotes metastasis progression [150]. As adrenosterone treatment downregulates Snail and Slug and recovered E-cadherin expression [150], this suggests that adrenosterone signaling may integrate Xmrk via the Ras/Raf/MEK/MAPK pathway, which can act through Snail and Slug to inhibit E-cadherin (Figure 2). However, the precise role of Xmrk in rabeprazole and olmesartan signaling is not currently understood. Evidently, various fish cell and transgenic-based models now exist that can provide a successful platform for investigating the role of Xmrk in future translational drug research.

## 6. Conclusions

Fish Xmrk models may be of value for studying human EGFR biology and related pathologies based on the degree of genetic and structural homology between the Xmrk receptor and the full-length EGFR isoform. Nonetheless, as these two receptors encode distinct nucleotide and protein sequences, there could be significant differences between Xmrk and human EGFR functions between these species. In addition, the mechanisms regulating Xmrk gene transcription, including its potential epigenetic regulation, are not well understood and could be distinct from those used in regulating EGFR gene transcription. Despite these potential differences between the Xmrk receptor and the EGFR, many genetic and functional characteristics of Xmrk have now been established using fish-derived cell lines, embryo and transgenic models, and Xmrk/EGFR chimeras, all of which have facilitated comparative analyses of Xmrk and EGFR oncogenic signaling. Although the initial focus in fish models was restricted to melanoma because of its historical association with Xmrk, more recent studies of HCC using transgenic zebrafish demonstrate there remain many unexplored avenues for studying the role of the EGFR and its related signaling networks in cancer and human pathologies. One application, minimally explored to date, uses high-throughput Xmrk translational drug discovery models for targeting human diseases that incorporate EGFR signaling. We conclude that fish-based models of Xmrk are well-positioned to become a future focus for studying EGFR.

## Figures and Tables

**Figure 1 cells-10-01132-f001:**
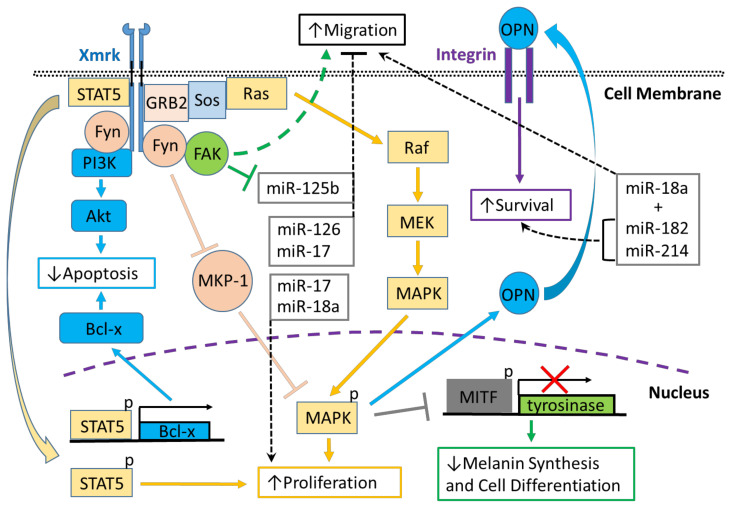
Xmrk signaling in the *Xiphophorus* melanoma model. Xiphophorus melanoma receptor kinase (Xmrk) is a receptor tyrosine kinase that recruits and activates several kinases, transcription factors and adaptor proteins integrated into signaling networks to regulate pigment cell cancer transformation. Activation of the kinase, phosphoinositide 3-kinase (PI3K), prevents apoptosis in malignant cells, as does activation of the transcription factor, signal transducer and activator of transcription 5 (STAT5), via B-cell lymphoma (Bcl-x) signaling, and STAT5 also promotes proliferation. The Src kinase protein, Fyn, acts as a docking protein and, through focal adhesion kinase (FAK), can induce pigment cell migration. Further, Fyn inhibits MAPK phosphatase 1 (MKP-1), a repressor of MAPK, causing the promotion of MAPK’s function to increase melanoma cell proliferation. Xmrk can also signal via the adaptor proteins, GRB2 and Sos, to activate the Ras/Raf/MEK/MAPK pathway, thereby enhancing proliferation. Additionally, MAPK signaling causes degradation of the transcription factor, microphthalmia transcription factor (MITF), leading to reduced transcription of tyrosinase, which decreases melanin synthesis and inhibits pigment cell differentiation. Another function of activated MAPK is to induce the synthesis of osteopontin (OPN), which is secreted, binds to integrins on the cell membrane surface, and promotes pigment cell survival. Several miRNAs that are regulated in fish Xmrk models also have functions in human melanoma. The miRNAs miR-18a, -182, and -214, can act to promote migration in human melanoma, while miR-182 and -214 can also stimulate cell survival. MiR-17, 125b, and -126 act to inhibit melanoma cell migration, with -125b’s action inhibited by FAK. MiR-17, along with miR-18a, also function to increase melanoma proliferation.

**Figure 2 cells-10-01132-f002:**
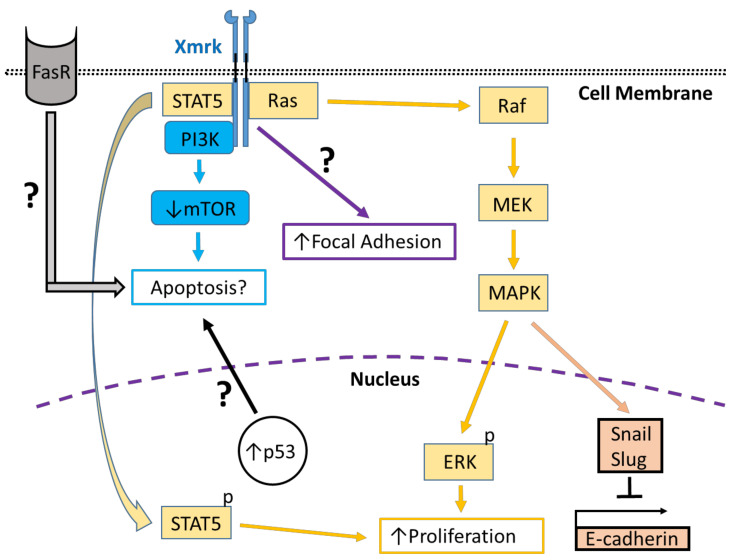
Xmrk signaling in the zebrafish hepatocellular carcinoma model. In transgenic zebrafish models of HCC, Xmrk signals through kinases and transcription factors that modulate liver cancer progression. PI3K activation may modulate apoptosis through the mechanistic target of rapamycin (mTOR), although mTOR expression may be decreased in HCC, making its effect on apoptosis uncertain. STAT5 signaling can promote proliferation, as does MAPK via activation of extracellular signal-regulated kinases (ERK). MAPK also has a dual role in signaling through the transcription factors, Snail and Slug, to inhibit E-cadherin activity. The formation of focal adhesions is increased through a pathway that is currently uncharacterized. Xmrk is also associated with increased expression of the tumor suppressor, p53, and the Fas receptor (FasR), which regulates apoptosis during HCC, but the details of their associated signaling mechanisms are not currently understood.

**Table 1 cells-10-01132-t001:** List of zebrafish EGFR isoforms in the Ensembl database. ID, transcript name, transcript length, protein name and protein length for all zebrafish EGFR isoforms listed in the Ensembl database.

Ensembl ID	Transcript	Length (nt)	Protein (Uniprot)	Length (aa)
egfra-201	ENSDART00000108964.5	1868	F1RBY7	503
egfra-202	ENSDART00000128514.2	1881	F1RA48	389
egfra-203	ENSDART00000136906.3	2437	F1R671	760
egfra-205	ENSDART00000147261.3	3168	F1QU74	243
egfra-206	ENSDART00000150499.3	3007	F1Q7X2	625
egfra-207	ENSDART00000164152.3	6151	A0A0R4IFV9	1191

**Table 2 cells-10-01132-t002:** List of human EGFR isoforms in the Ensembl and NCBI genomic databases. ID, transcript name, transcript length, protein name and protein length for all human EGFR isoforms listed in the Ensembl (top) and NCBI (bottom) databases. Isoforms highlighted in green are common to both databases.

Ensembl ID	Transcript	Length (nt)	Protein (Uniprot)	Length (aa)
EGFR-201	ENST00000275493.7	9905	P0053-1	1210
EGFR-202	ENST00000342916.7	2239	P00533-4	628
EGFR-203	ENST00000344576.6	2864	P00533-3	705
EGFR-204	ENST00000420316.6	1570	P00533-2	405
EGFR-205	ENST00000450046.1	691	C9JYS6	128
EGFR-206	ENST00000454757.6	5464	E9PFD7	1165
EGFR-207	ENST00000455089.5	3844	Q504U8	1091
**NCBI ID**	**Transcript**	**Length (nt)**	**Protein (NCBI)**	**Length (aa)**
Isoform A	NM_005228.5	9905	NP_005219.1	1210
Isoform G	NM_001346899.2	9770	NP_001346899.2	1165
Isoform I	NM_001346941.2	9104	NP_001333870.1	943
Isoform F	NM_001346898.2	3983	NP_001333827.1	1136
Isoform E	NM_001346897.2	3848	NP_001333826.1	1091
Isoform D	NM_201284.2	2872	NP_958441.1	705
Isoform B	NM_201282.2	2254	NP_958439.1	628
Isoform C	NM_201283.2	1575	NP_958440.1	405
Isoform H	NM_001346900.2	9676	NP_001333829.1	1157

## Data Availability

Not applicable.

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
