# Peer review of "Xmrks the Spot: Fish Models for Investigating Epidermal Growth Factor Receptor Signaling in Cancer Research"

_cells, 2021, doi:10.3390/cells10051132_

Round 1

Reviewer 1 Report

This review gave an overview on the roles of Fish Xmrk models for studying human EGFR biology and related pathologies. Overall it was well written and requires minimal content / organizational changes. Comments: 1. In the manuscript, the authors reviewed the genetic characterization and comparison of Xmrk with the human EGFR and Xmrk in Fish models of melanoma and hepatocellular Carcinoma. However, the abstract seems like describe the current situation but not summarize the main points of the manuscript. It should be revised accordingly. 2. Since there are several microRNA are involved in Xmrk signaling, it is highly recommend adding the microRNA in the Figure to make it easier to grasp. 3. The authors should discuss the therapeutic agents targeting Xmrk signaling. The author should discuss this briefly.

Reviewer 2 Report

The abbreviation is different in the figure and in the text. Shouldn't it be unified?
For example, STAT5 in text  and Stat5 in figure.

Author Response

The abbreviation is different in the figure and in the text. Shouldn't it be unified?
For example, STAT5 in text and Stat5 in figure.

Our response to the reviewer comment: As requested by the reviewer, we have introduced uniform spelling for all abbreviations that reconciles the text and the figures.

Round 2

Reviewer 1 Report

The authors have satisfactorily responded to the comments that the subject matter of this work is acceptable for publication.